# Molecular Epidemiology of Respiratory Syncytial Virus during 2019–2022 and Surviving Genotypes after the COVID-19 Pandemic in Japan

**DOI:** 10.3390/v15122382

**Published:** 2023-12-04

**Authors:** Sayaka Yoshioka, Wint Wint Phyu, Keita Wagatsuma, Takao Nagai, Yasuko Sano, Kiyosu Taniguchi, Nobuo Nagata, Kazuhiko Tomimoto, Isamu Sato, Harumi Kaji, Ken Sugata, Katsumi Sugiura, Naruo Saito, Satoshi Aoki, Eitaro Suzuki, Yasushi Shimada, Hirotsune Hamabata, Irina Chon, Teruhime Otoguro, Hisami Watanabe, Reiko Saito

**Affiliations:** 1Division of International Health (Public Health), Graduate School of Medical and Dental Sciences, Niigata University, Niigata 951-8510, Japan; syoshioka@med.niigata-u.ac.jp (S.Y.); amaramyat2018@gmail.com (W.W.P.); waga@med.niigata-u.ac.jp (K.W.); irinachon@med.niigata-u.ac.jp (I.C.); 2Infectious Diseases Research Center of Niigata University (IDRC), Niigata University, Niigata 951-8510, Japan; totoguro@med.niigata-u.ac.jp (T.O.); hwatanabe@med.niigata-u.ac.jp (H.W.); 3University of Medicine, Yangon, Myanmar; 4Nagai Pediatric Clinic, Takamatsu 760-0002, Japan; t-nagai@me.pikara.ne.jp; 5Sano Clinic, Niigata 950-0853, Japan; yassano@agate.plala.or.jp; 6Mie National Hospital, Tsu 514-0125, Japan; tngk7g04@gmail.com (K.T.); osugisugisugi626@yahoo.co.jp (K.S.); 7Hiraoka-Kouen Pediatric Clinic, Sapporo 004-0872, Japan; nnagata@wine.ocn.ne.jp; 8Tomimoto Pediatric Clinic, Hachinohe 031-0823, Japan; altair@eagle.ocn.ne.jp; 9Yoiko Pediatric Clinic Sato, Niigata 950-0983, Japan; e-yoiko@guitar.ocn.ne.jp; 10Kaji Clinic, Hachioji 193-0816, Japan; harumikaji@k4.dion.ne.jp; 11Saito Pediatric Clinic, Moriyama 524-0022, Japan; doctor@children.or.jp; 12Aoki Pediatrics, Kitakatsuragi 636-0002, Japan; aoki-sy@af.em-net.ne.jp; 13Suzuki Pediatric Clinic, Ube 755-0151, Japan; 14Shimada Pediatrics, Kamiamakusa 869-3601, Japan; shimaday@ac.mbn.or.jp; 15Awase Daiichi Clinic, Okinawa 904-2172, Japan; awase555@cello.ocn.ne.jp

**Keywords:** respiratory syncytial virus, COVID-19 pandemic, molecular epidemiology, G gene, phylogeography

## Abstract

To evaluate the changes in respiratory syncytial virus (RSV) collected between 2019 and 2022, we analyzed RSV-A and RSV-B strains from various prefectures in Japan before and after the COVID-19 pandemic. RT-PCR-positive samples collected from children with rapid test positivity at outpatient clinics in 11 prefectures in Japan were sequenced for the ectodomain of the G gene to determine the genotype. Time-aware phylogeographic analyses were performed using the second hypervariable region (HVR) of the G gene from 2012 to 2022. Of 967 samples, 739 (76.4%) were found to be RSV-positive using RT-PCR. RSV peaked in September 2019 but was not detected in 2020, except in Okinawa. Nationwide epidemics occurred with peaks in July 2021 and 2022. The genotype remained the same, ON1 for RSV-A and BA9 for RSV-B during 2019–2022. Phylogeographic analysis of HVR revealed that at least seven clusters of RSV-A had circulated previously but decreased to two clusters after the pandemic, whereas RSV-B had a single monophyletic cluster over the 10 years. Both RSV-A and RSV-B were transferred from Okinawa into other prefectures after the pandemic. The RSV epidemic was suppressed due to pandemic restrictions; however, pre-pandemic genotypes spread nationwide after the pandemic.

## 1. Introduction

Respiratory syncytial virus (RSV) causes severe lower respiratory tract infections in infants and small children, and half the patients are infected within the first year of life [1]. It is estimated that RSV caused 33.0 million acute lower respiratory infection episodes, 3.6 million acute lower respiratory infection hospitalizations, and 101,400 overall deaths in children aged 0–60 months worldwide in 2019 [2]. RSV can recur throughout life, and elderly and immunocompromised adults are at risk of serious infections [3]. As for vaccines against RSV, in May 2023, the U.S. Food and Drug Administration approved the two RSV vaccines for use in the U.S. exclusively in persons aged 60 years or older [4,5]. In August 2023, the Ministry of Health, Labour, and Welfare, Tokyo, Japan approved the RSV vaccine for use in Japan exclusively in persons aged 60 years or older [6]. In addition, recombinant vectors, subunits, attenuated live vectors, and nucleic acid vaccines for maternal and child immunity, respectively, are in development [7].

RSV belongs to the genus Orthopneumovirus of the family Pneumoviridae, and is a single-stranded negative-sense ribonucleic acid (RNA) virus with an envelope [1]. Its genome is approximately 15 kb in length and encodes 11 proteins. Of these, glycoprotein (G) targets ciliated respiratory epithelial cells and is considered a major antigen [8]. Its genetic sequence is the most variable among proteins of RSV [9], which is considered to be due to the selection pressure exerted by the host’s immune system [10]. RSV is divided into two subtypes (RSV-A and RSV-B) [11]: RSV-A has been classified into genotypes GA1–7 [12], SAA1 [13], NA1–4 [14,15], and ON1 [16], and RSV-B into genotypes GB1–5 [12], SAB1–4 [13], URU1–2 [17], and BA1–10 [18], using the classification of the second hypervariable region (HVR2) of the G gene, which spans from 270 to 342 bp. Currently, the most frequently detected genotypes in the world are ON1 for RSV-A and BA9 for RSV-B [19]. Genotype ON1 with a 72-nucleotide duplication in HVR2 was identified in Canada in 2010 [16]. ON1 evolved from NA1, which we previously reported as a new genotype in 2004–2005, and rapidly spread to many countries after 2010 [14]. BA genotype, with a 60-nucleotide duplication in HVR2, was first reported in Buenos Aires in 1999 and became the predominant RSV-B strain [20]. Recently, it was proposed that the G ectodomain region, approximately 800 bp long, which includes HVR2, is the minimum region suitable for genotyping [21].

RSV epidemics are typically observed in the fall and winter in countries located in the temperate zone of the Northern Hemisphere [22,23]. In Japan, RSV was prevalent in fall and winter from 2007 to 2015, peaking in November-December, whereas from 2016 to 2019, the seasonality shifted to late summer and early fall, peaking in September–October [24,25].

For more than 20 years, our department has conducted a series of investigations on the molecular epidemiology of RSV in outpatient clinics in Japan. Through our investigations, we reported several new genotypes of NA1, NA2, BA7, BA8, BA9, and BA10, and found that a newly reported genotype in Canada, ON1, was imported to Japan in 2013 [14,18,26]. In addition to molecular epidemiology, we reported that RSV seasonality in Japan moves from winter to summer [24] and another report on clinical virology showed that patient age contributes to symptom duration and viral reduction in outpatients [27]. This study was also conducted as a part of this investigation, and samples were collected from various prefectures in Japan through our own network.

COVID-19 has caused a pandemic worldwide since December 2019 [28]. Non-pharmaceutical interventions (NPIs) such as travel restrictions, school closures, and social distancing, which were implemented to suppress the COVID-19 pandemic, have changed the pattern of RSV circulation [25,29,30]. Many countries reported a decrease in RSV cases in 2020 and an out-of-season epidemic in 2021 [22,23,31,32,33,34]. Several research teams have published reports on the timing of the epidemic in Japan following the COVID-19 pandemic [35,36,37]. So far, limited groups reported the genotypes of RSV restricted to single locations. [38,39]. In this study, we investigated the subtypes and genotypes of RSV using the G ectodomain region of samples collected from various prefectures in Japan before and after the COVID-19 pandemic. In addition, we performed a longitudinal phylogeographical analysis of RSV collected from 2012 to 2022 using the HVR2 region of the G gene to clarify the changes in genotypes associated with the pandemic.

## 2. Materials and Methods

### 2.1. Study Population and Clinical Samples

This observational study was conducted at 11 outpatient clinics located in 11 different prefectures (i.e., Hokkaido, Aomori, Tokyo, Niigata, Mie, Shiga, Nara, Kagawa, Yamaguchi, Kumamoto, and Okinawa Prefecture) in Japan over an approximately three-year period from March 2019 to December 2022, including the COVID-19 pandemic period. Healthy children under six years of age with symptoms of suspected RSV infection such as fever (≥37.5 °C), cough, or rhinorrhea were screened using the Quick-Navi^TM^ Flu + RSV rapid test kit (RDT) (Denka, Tokyo, Japan) or the Quick-Navi^TM^ RSV2 rapid test kit (Denka, Tokyo, Japan). RDTs are approved for diagnostic purposes in Japan and are covered by the National Health Insurance scheme. The sensitivity and specificity of these RDTs in nasal swabs are reported to be 98.3% and 98.8%, respectively, for Quick-Navi^TM^ Flu + RSV [40] and 97.4% and 96.4% for Quick-Navi^TM^ RSV2 [41]. Nasal swabs, nasopharyngeal aspirates, rhinorrhea, and saliva were collected by clinicians after written informed consent was obtained from the parents or guardians of the patients. Patient demographics and clinical information, such as age, sex (male or female), clinical symptoms (fever, cough, or rhinorrhea), date of clinic visit, number of weeks of pregnancy, and underlying diseases (congenital heart disease, chronic respiratory disease, Down syndrome, or immunodeficiency) were recorded. A child was considered premature only if he or she was 28 weeks or less in gestation and less than 12 months old, or 29–35 weeks of gestation and less than 6 months of age at the first clinical visit. Specimens were frozen at −20 °C, transported to the Division of International Health (Public Health), Graduate School of Medical and Dental Sciences, Niigata University, Niigata, Japan, and stored at −80 °C for further virologic examination as follows. This study was approved by the Niigata University Ethics Review Committee (approval number 2021-0190).

### 2.2. Subtyping

Viral RNA was extracted from 140 μL of clinical specimens using the QIAamp Viral RNA Mini test Kit (QIAGEN, Hiden, Germany) according to the manufacturer’s instructions. Reverse transcription was performed to create complementary deoxyribonucleic acid (cDNA) incubating with random primers and Moloney murine leukemia virus reverse transcriptase (Invitrogen Corporation, Carlsbad, CA, USA) at 37 °C for 1 h [26].

A TaqMan probe real-time polymerase chain reaction (RT-PCR) assay was performed for subtyping (RSV-A or -B) using probe sets targeting the membrane fusion (F) protein of RSV as reported previously [26]. The positive Ct values were defined as ≤40 cycles and the curves over 40 were considered as negative.

### 2.3. National RSV Surveillance Data

To support the epidemic trend of RSV infections, we used the number of RSV outpatient visits per sentinel medical institution in each prefecture in Japan from the Infectious Disease Weekly Report (IDWR), sourced from the National Epidemiological Surveillance of Infectious Diseases (NESID) data published by the National Institute of Infectious Diseases (NIID), Tokyo, Japan [24,25,26]. In the NESID system, approximately 3000 pediatric outpatient sentinels (hospitals and clinics) report the weekly number of patients diagnosed with RSV infection to the public health centers in each prefecture. The number of sentinels assigned to each public health service area is determined based on population size: a public health center with <30,000 individuals has one sentinel, a center with 30,000–75,000 individuals has two sentinels, and one with >75,000 individuals has ≥3 sentinels, as determined by the following formula: 3 + (population − 75,000)/50,000 [24]. RSV cases were defined by viral detection using rapid diagnostic kits, viral isolation, or antibody elevation in paired sera according to the guidelines of the Ministry of Health, Labour and Welfare, Tokyo, Japan (https://www.mhlw.go.jp/bunya/kenkou/kekkaku-kansenshou11/01-05-15.html (accessed on 5 September 2023)). These sentinel sites forward clinical data to approximately 60 prefectural or municipal public health sectors, which are electronically reported to the NIID. The number of RSV cases was released weekly via a website. The monthly number of RSV cases per sentinel site in the NESID during the study period was compiled based on weekly data (https://www.niid.go.jp/niid/ja/idwr.html (accessed on 5 September 2023)).

### 2.4. The G Ectodomain Region Sequencing

After subgrouping by RT-PCR, positive samples underwent conventional PCR, targeting the glycoprotein (G) ectodomain region using the PrimeScript™ II High Fidelity One Step RT-PCR Kit (TaKaRa Bio Inc., Shiga, Japan) and the following primers: RSVBG10f-TAG (5′-CACGACGTTGTAAAACGACCGCAATGATAATCTCAACCTC-3′) and RSVBF1r-TAG (5′-CAGGAAACAGCTATGACCCAACTCCATKGTTATTTGCC-3′, with K = G or T); underlined parts indicate the universal M13 tag as reported by Goya et al. [42]. Amplicons were 843 bp for RSV-A and RSV-B. A total of 3.0 μL of viral RNA was added to the 17 μL reaction mixture. The components of the reaction mixture were as follows: 10 µL of 2 × One step High Fidelity Buffer, 0.4 μL of PrimeScript™ II RT Enzyme Mix, 1.6 μL of PrimeStar GXL for 1step RT-PCR, 1.6 μL of 10 pmol/μL RSVBG10f-TAG primer, 1.6 μL of 10 pmol/μL RSVBF1r-TAG primer, and 1.8 μL of nuclease-free water. Thermal cycling conditions were 45 °C for 10 min, 94 °C for 2 min, followed by 35 cycles of 98 °C for 10 s, 62 °C for 15 s, and 68 °C for 30 s. Amplified PCR products were purified using a QIAquick PCR purification kit (QIAGEN, Hiden, Germany), labeled with a BigDye terminator cycle sequencing kit version 3.1 (Applied Biosystems, Pleasanton, CA, USA) according to the manufacturer’s instructions, and analyzed on an ABI Prism 3130xl Genetic Analyzer (ThermoFisher Scientific Inc., Waltham, MA, USA) or a SeqStudio™ Genetic Analyzer (ThermoFisher Scientific Inc., Waltham, MA, USA). Sequencing primers used for RSV-A were as follows: RSVBG10f-TAG, RSVBF1r-TAG, RSVG2AF (5′-GAAGTGTTCAACTTTGTACC-3′), RSVG1AR (5′-GGTTTTTTGTTGGGTATTCTTTTGC-3′), RSVG2AFnew (5′-GAAGTGTTCAAYTTTGTRCC-3′, with Y = C or T and R = A or G), and RSVG1ARnew (5′-GGTTTTTTGTTMGGTATTCTRT-3′, with M = A or C and R = A or G). RSVG1AR and RSVG2AF primers were used for samples collected in 2019 and 2020, and RSVG1ARnew and RSVG2AFnew primers were used for samples collected in 2021 and 2022, respectively. Sequence primers used for RSV-B were as follows: RSVBG10f-TAG, RSVBF1r-TAG, RSVGPB (5′-AAGATGATTACCATTTTGAAGT-3′), and RSVG2BF-R(C) (5′-TTGTGCTGTTGTATGGTGTG-3′).

For samples collected from 2012 to 2018, RT-PCR-positive samples underwent conventional PCR, targeting the HVR2 of the G gene [26], a narrower region than that conducted during 2019–2022. The amplified PCR products were purified using a QIAquick PCR purification kit (QIAGEN, Hilden, Germany), labeled with a BigDye Terminator Cycle Sequencing kit version 3.1 (Applied Biosystems, Pleasanton, CA, USA) according to the manufacturer’s instructions, and analyzed using an ABI Prism 3130xl Genetic Analyzer (Thermo Fisher Scientific Inc., Waltham, MA, USA). PCR primers were used for sequencing.

The obtained sequences were assembled using the Lasergene SeqMan Pro package version 17.3.0 (DNASTAR, Madison, WI, USA) and then aligned using ClustalW in BioEdit software version 7.2.6.1 (https://bioedit.software.informer.com/7.2/ (accessed 5 September 2023)).

### 2.5. Nucleotide Sequences Accession Numbers in the GISAID

The G ectodomain nucleotide sequences of RSV-A (208 strains) and RSV-B (102 strains) were registered in the GISAID database under the accession numbers EPI_ISL_15953175 to EPI_ISL_18487961 (Appendix A). The HVR2 nucleotide sequences of RSV-A (113 strains) and RSV-B (95 strains) collected from 2015 to 2018 were registered in the GISAID database under the accession numbers EPI_ISL_18396054 to EPI_ISL_18406067.

### 2.6. Phylogenetic and Molecular Evolutionary Analysis Using Maximum Likelihood (ML) and Bayesian Markov Chain Monte Carlo (MCMC) Methods

To determine the genotype and examine the genetic relationship between the strains obtained in this study and those circulating worldwide, we first aligned our G ectodomain sequences with representative sequences of the genotype/subgenotype/lineages, as defined by Goya et al. [21]. Strains from Japan or other countries that had at least 98% similarity to the sequences obtained in this study were searched for using the Basic Local Alignment Search Tool (BLAST) and downloaded from GenBank to add to multiple alignment [43]. We then performed a phylogenetic tree analysis of this alignment, including the query sequence and reference strain, using two methods: the maximum likelihood (ML) method and Bayesian inference. A bootstrap of 80% or greater obtained from the ML method and a posterior probability value of 0.8 or greater obtained from Bayesian estimation were considered to have high statistical support. In addition, clusters with three or more non-identical sequences from different countries, different occurrences, or a combination of both and with high statistical support were considered monophyletic [21].

An ML tree was constructed using MEGA version 6 software [44] with 1000 bootstrap iterations. GTR + G was selected as the best-fit nucleotide substitution model for RSV-A and RSV-B.

A Bayesian tree was constructed using the Bayesian Markov Chain Monte Carlo (MCMC) method in the BEAST package version 1.10.4 [45]. Correlations between sampling dates and genetic variation over time were calculated using the TempEst software version 1.5.3 (http://tree.bio.ed.ac.uk/software/tempest/ (accessed 5 September 2023)) [46]. As the best-fit nucleotide substitution model, GTR + G for RSV-A and RSV-B was selected and analyzed under an uncorrelated relaxed clock model or strict clock model; MCMC chains were run every 50 million iterations, with subsampling every 10,000 iterations. Prior evolution rates of 3.63 × 10^−3^ substitutions/site/year (95% highest posterior density [HPD], 2.31 × 10^−3^ to 4.95 × 10^−3^) for RSV-A and 4.56 × 10^−3^ substitutions/site/year (95% HPD, 3.22 × 10^−3^ to 6.03 × 10^−3^) for RSV-B [47] were used in this analysis to calibrate the molecular clock. Convergence was evaluated by Tracer software version 1.7.2 (https://github.com/beast-dev/tracer/releases/tag/v1.7.2/ (accessed 5 September 2023)) and accepted when the effective sample size (ESS) value for each parameter ≥200 [48]. The maximum clade credibility (MCC) tree was summarized using Tree Annotator version 1.10.4, with 10% burn-in discarded, and visualized using FigTree software version 1.4.4 (https://github.com/rambaut/figtree/releases/ (accessed 5 September 2023)).

Furthermore, to estimate a longitudinal molecular evolutionary pathway of RSV in Japan, we performed a time-aware MCMC phylogenetic and molecular evolutionary analysis by the BEAST software, using the shorter HVR2 region by adding the sequence data of the HVR2 of strains for RSV-A and RSV-B collected from 2012 to 2018 in our past study. The reason for the HVR2 analysis was that we had been sequencing this region for a long time, so we thought we could clarify and support the shift in genotypes before and after COVID-19. For this dataset, the best-fit nucleotide substitution model for RSV-A and RSV-B, TN93 + G, was selected and analyzed using an uncorrelated relaxed clock model or a strict clock model. The MCMC chains were run every 250 million iterations for RSV-A and 100 million iterations for RSV-B with subsampling every 10,000 iterations.

### 2.7. Statistical Analysis

Data are described as median (interquartile range [IQR]) for continuous variables and frequency (%) for categorical variables. Fisher’s exact test was used to compare proportions. All analyses were performed using Excel 2019 and EZR software version 1.60 [49], and a *p* value of <0.05 was considered a significant difference.

## 3. Results

### 3.1. Laboratory Detection and Clinical Characteristics of Patients in This Study

A total of 967 pediatric outpatients were enrolled between March 2019 and December 2022. Of the 967 samples collected, 739 of 967 cases (76.4%) were RSV-positive to RT-PCR. The clinical information of pediatric patients identified by RT-PCR is presented in Table 1. During the entire study period, the median [IQR] age of patients was 1.4 [0.8–2.2] years, with 503 (68.1%) patients from 0 to <2 years old and 236 (31.9%) patients from 2 to <6 years old (Table 1). Of them, 394 (53.3%) were male patients and 345 (46.7%) were female. Among the symptoms, the most prevalent was cough, presented by 720 (97.4%), followed by rhinorrhea (709 (95.9%)) and fever (603 patients (81.6%)) (Table 1). More than 80% of patients presented with at least one of these symptoms. The percentage of patients presenting with rhinorrhea was significantly lower in 2021 (*p* = 0.027) and 2022 (*p* < 0.001) than in 2019 and was also significantly lower in 2022 than in 2021 (*p* = 0.045). The reason for the significant decrease in rhinorrhea in this study remains unknown, but we will continue to observe whether this trend persists. The number of pediatric patients with premature births or underlying diseases was 51 (6.9%). Statistical analysis revealed significant differences in the percentages of patients in each age group (*p* < 0.001). Specifically, the percentage of patients aged 2 to <6 years versus those aged 0 to <2 years increased significantly in 2021 compared to that in 2019 (*p* < 0.001) and 2020 (*p* < 0.001), indicating that older children were infected with RSV after 2021 (Table 1).

### 3.2. Epidemic of RSV from 2019 to 2022 in Japan

According to national surveillance data (NESID data), the RSV epidemic in 2019 at the national level in Japan began in July, peaked in September, and converged in November (Figure 1), as observed in 2016–2018 [25]. In 2020, the number of patients with RSV was low, and no nationwide peak was observed. In 2021, the RSV epidemic began in April, peaked in July, and converged in October, with a peak timing three months earlier than the pre-pandemic period. In 2022, the epidemic began in June, peaked in July, and ended in December, similar to that in 2021. Compared with the size of the RSV epidemic, the number of patients visiting per sentinel in the peak month was 12.48 in 2019, but increased to 21.49 in 2021 after the pandemic, approximately 1.7 times larger. In 2022, the number of patients visiting the peak month was 8.87, 0.7 times smaller than that in 2019, showing a large surge in 2021 after the COVID-19 pandemic. Since the national surveillance data did not differentiate between the RSV subgroups, we compared the RSV subtypes detected by RT-PCR in this study (Figure 1). Our investigation showed that in 2019, the epidemic was a mixed circulation of RSV-A and -B with a predominance of RSV-A and peaks coinciding with the NESID data in September. In 2020, RSV-B was detected from October to December owing to local circulation in the study area of Okinawa. In 2021, the epidemic was again a mixture of RSV-A and -B dominated by RSV-A. However, our sample data showed double peaks in May and October, which differ from the NESID data for 2021. In 2022, the epidemic was also dominated by RSV-A, with our data showing a peak in September, later than that of NESID.

The breakdown of the 11 prefectures that participated in this study was also analyzed (Table 2, Figure 2). During the study period, we observed that epidemics in Okinawa, the southernmost island, occurred at somewhat different times than those in other prefectures. In 2019, RSV epidemics were observed in all prefectures except Okinawa at about the same time, from July to November. In Okinawa, the epidemic occurred from May to September, approximately two months earlier than in other prefectures. Our previous study from 2012 to 2015 also showed that the epidemic started earlier in Okinawa than in other areas [24,26]. In 2020, RSV was not detected in any of the prefectures that participated in this study, except Okinawa, which showed RSV-B circulation from September to December. In 2021, RSV was detected in prefectures other than Okinawa in spring and summer, and in Okinawa mainly in fall. The epidemic began in Kumamoto in January, which was the earliest in the study area. The epidemic peaked in Kumamoto in April; Nara in May; Aomori and Yamaguchi in June; Tokyo, Mie, and Shiga in July; and Hokkaido and Kagawa in August 2021. The epidemic in 2021 appears to have moved mainly from populated areas in the southwest to less populated areas in the northeast. In Okinawa, a peak was observed in October in 2021, similar to that in 2020. There was a difference in the subgroups circulated by prefecture; RSV-A was dominant in Nara, Yamaguchi, and Okinawa, whereas RSV-B was dominant in Tokyo, Mie, Shiga, and Kumamoto, and both RSV-A and RSV-B were equally circulated in Hokkaido in 2021. In 2022, the epidemic occurred from June to December in all prefectures except Okinawa. The epidemic peaked in Mie and Nara in July; Shiga in August; Tokyo, Kagawa, and Yamaguchi in September; Hokkaido and Aomori in October; and Niigata and Kumamoto in November 2022. The 2022 epidemic again seemingly moved mainly from the populated prefectures in the southwest to the less populated prefectures in the northeast. In Okinawa, the epidemic peaked in October, as in the previous two years.

### 3.3. RSV Genotype

The G protein ectodomain region was successfully sequenced in 208 of 503 (41.4%) RSV-A and 102 of 236 (43.2%) RSV-B. After excluding identical sequences, ML and MCC trees were created for 92 RSV-A and 54 RSV-B strains. The ML tree is shown in Figure 3, and the MCC tree is shown in Appendix A. For RSV-A, the ML and MCC trees indicated that all strains collected from 2019 to 2022 belonged to the same lineage as defined by Goya et al. [21] (Figure 3A and Appendix A). All RSV-A strains had a 72 bp nucleotide duplication in the C-terminal region of the G gene and were classified as genotype ON1 [16]. Our RSV-A strains had five amino acid mutations collected in 2019–2022, T113I, V131D, N178G, H258Q, and H266L, compared to the reference strain China/LZ170020467/17 (accession number MH290724). After 2021, two amino acid mutations showed: D214Y and G272S. The strains that showed high similarity to RSV-A were from China, the Philippines, Thailand, Russia, the United Kingdom, France, Austria, the Netherlands, Kenya, the United States, Argentina, Australia, and New Zealand. For RSV-B from the ML and MCC trees, all strains collected from 2019 to 2022 belonged to the same lineage, and all RSV-B strains had a duplicated base of 60 bp in the HVR2 of the G gene and were classified as genotype BA9 [18,21] (Figure 3B and Appendix A). RSV-B collected from 2020 to 2022 had the H128R amino acid mutation compared to the strains collected in 2019. The RSV-B strains collected in this study from 2019 to 2022 are similar to those collected in the Philippines, India, Russia, the United Kingdom, Spain, Switzerland, Austria, the United States, and Australia.

### 3.4. Time Scale Phylogenetic Tree of RSV Using the Bayesian MCMC Method

To estimate the longitudinal geographic transmission of RSV-A and -B in Japan, we used our previously sequenced HVR2 data collected during 2012–2018 in addition to the ectodomain sequences from 2019 to 2022 from this study. Time-aware MCC trees were created for the HVR2 of G gene using the Bayesian MCMC method in the BEAST software (Figure 4). For RSV-A, MCC trees were constructed for 273 strains, including the 92 strains used in the ML tree construction from 2019 to 2022, plus 181 strains collected in our previous study from 2012 to 2018 (Figure 4A). As reported in a previous study, only genotype NA1 was detected in 2012, ON1 was first detected in 2013, and since 2014, it has replaced NA1 [26]. ON1 strains collected between 2013 and 2019 branched into multiple clusters (numbered 1 to 7), each presumed to be derived from strains collected at various prefectures, including Tokyo, Niigata, Mie, Shiga, and Kumamoto, with each cluster circulating in various prefectures. However, since 2020, when the COVID-19 pandemic occurred, only three clusters (1, 6, and 7) remained by 2021. Clusters 1 and 6 were estimated to be derived from strains in Okinawa, and cluster 7 was derived from Yamaguchi (southwest Japan). However, only two clusters originating from Okinawa, 1 and 6, remained in 2022. For RSV-B, MCC trees were constructed for 166 strains, including the 54 strains used in the ML tree construction from 2019 to 2022, plus 112 strains collected in our previous study from 2013 to 2018 (Figure 4B). In contrast to RSV-A, RSV-B forms a monophyletic cluster. In the MCC tree, the most probable root was the strain collected from Kumamoto in 2013. The MCC tree suggested that RSV-B evolved from strains collected in southwestern Japan (Nara, Yamaguchi, Kumamoto, and Okinawa) or from the densely populated Tokyo area before the COVID-19 pandemic. After the pandemic, strains originating from Okinawa survived and spread to other areas of Japan in 2021 and 2022.

## 4. Discussion

In this study, we evaluated the changes in epidemic timing, subtypes, and genotypes of RSV before and after the COVID-19 pandemic in samples collected from 11 different areas of Japan. During the study period, an RSV epidemic was observed from summer to fall in 2019. However, it became silent in 2020, except in Okinawa, when the COVID-19 pandemic began. A large epidemic occurred from spring to summer in 2021, and a small epidemic occurred from summer to fall in 2022. The most common genotypes were ON1 for RSV-A and BA9 for RSV-B from 2019 to 2022, and the genotypes did not change before and after the pandemic. A time-aware MCC tree analysis of HVR2 in the G gene from 2012 to 2022 revealed that multiple clusters of RSV-A circulated in the studied areas of Japan; however, the diversity decreased after the COVID-19 pandemic. RSV-B formed a monophyletic lineage. Notably, after the pandemic, both RSV-A and RSV-B strains were seeded from Okinawa.

In this study, the proportion of RSV patients aged 2 to <6 years in 2021 was significantly higher than that in 2019 and 2020, indicating that older children were infected after the COVID-19 pandemic. Significant increases in the proportion of patients aged 2 years and older in 2021 compared with before the COVID-19 have also been reported not only in previous studies conducted in Japan [36,39], but also in Taiwan, France, the United Kingdom, and Argentina [22,32,33,34]. In Japan, we reported that the RSV epidemic had almost disappeared by 2020 because of various non-pharmaceutical measures against COVID-19, such as the hygiene measures and travel restrictions implemented in 2020 [25]. Thus, lack of exposure to RSV may reduce immunity in children. Reicherz et al. reported that antibody titers in samples collected from infants in 2021 were 15-fold lower than those collected in 2020 [50]. Therefore, the increase in the number of patients aged 2 to <6 years in 2021 was associated with low antibody levels in this age group during the COVID-19 measurements, and rebound infections occurred in 2021.

Along with the change in age groups, there was a nationwide shift in the epidemic season in Japan in 2021, resulting in a substantially large outbreak in July, earlier by two months compared to that observed before the COVID-19 pandemic [35,36,37,51]. Other countries, such as Taiwan, the United States, the United Kingdom, Argentina, and Australia, reported resurgences and a nationwide shift of epidemic season of RSV occurred in 2021 after the relaxation of non-pharmaceutical measurements [23,31,32,33,34]. The resurgences of out-of-season RSV can be attributed to a decline in population immunity resulting from an extended period of minimal RSV exposure, referred to as “RSV immunity debt” [52]. Since the factors influencing RSV seasonality and the processes through which RSV is annually reintroduced are not entirely understood, it is crucial to assess the effects of non-pharmaceutical interventions on RSV. We have previously reported that improved hygiene and sanitary measures and travel restrictions for COVID-19 in Japan can contribute to the suppression of RSV activity [25]. The factors contributing to the decline and resurgence of RSV activities, as well as the levels of RSV immunity, warrant thorough investigation. These insights are crucial for informing public health policymakers in devising effective strategies for the controls of respiratory virus infections.

We also focused on the shift in the epidemic timing among prefectures before and after the COVID-19 pandemic. From 2007 to 2019, Okinawa was the earliest prefecture where outbreaks were observed, almost 2 months before the other prefectures [24,26]. This was thought to be because Okinawa belongs to the subtropical zone, and it has often been reported that RSV cases increase in tropical or subtropical zones during the hot rainy season [24]. Interestingly, in 2020, during the first year of the COVID-19 pandemic, RSV-B cases were observed only in Okinawa in October and November; no cases were detected in other prefectures in our study. In fact, it is noteworthy that no RSV epidemic was observed in the United Kingdom or the United States [23,53], both of which are located in the northern hemisphere, while RSV-B continued to be prevalent in Okinawa, Japan. The reason for the continued prevalence of RSV-B in Okinawa could not be clarified in this analysis, but future detailed analysis at the prefectural level or in conjunction with actual data from other countries around the world may help to clarify the cause. In 2021, RSV-A and -B cases started to rise in Kumamoto, which is located on Kyushu Island in the southwestern part of Japan, and then moved from the southwest to northeast. Notably, densely populated areas such as Tokyo had an early rise in cases in 2021, even though they are located in northeastern Japan. In contrast, the epidemic in less populated areas, such as Kagawa on Shikoku Island, began late, even in the southwestern part. In 2022, the epidemic began in June in all prefectures except Okinawa, with Okinawa being the latest, starting in August. Again, the epidemic seemed to move from the populated southwest to less populated prefectures, mainly in the northwest. Rojo et al. found that new RSV outbreaks began in densely populated areas and subsequently spread to other areas in Argentina [42]. Thus, the timing of RSV outbreaks among Japanese prefectures may have been more influenced by population density, which caused more frequent human-to-human transmission than climatic conditions after the COVID-19 pandemic. Zheng et al. reported that, although a shift in the timing of RSV outbreaks was observed after the COVID-19 pandemic in the United States, the timing of outbreaks among states was similar to that before it [54]. The reason for the difference in the timing of RSV outbreaks among Japanese prefectures could be explained by the continued infection control measures compared with the United States; however, further attention should be paid to how the timing of outbreaks will change in Japan during the post-pandemic period.

ML phylogenetic tree analysis of the G ectodomain region revealed that the strains collected after the COVID-19 pandemic for both RSV-A and RSV-B belonged to the same lineage as the strains collected before. This suggests that the strains that were prevalent in 2019 were circulating in Japan even during a period when RSV was rarely detected. Regarding the strains prevalent since the COVID-19 pandemic, some research teams have reported that they belong to the same lineage as before the COVID-19 pandemic in China, Austria, Slovenia, Italy, and the United States [55,56,57,58,59], while Eden et al. reported the emergence of a new lineage in Australia [31]. Several characteristic amino acid mutations were detected in the RSV G gene sequences collected during the study period. In RSV-A, five amino acid mutations of T113I, V131D, N178G, H258Q, and H266L were detected from the strains collected in various parts of Japan from 2019 to 2022. Strains with these five mutations were detected in China from 2018 to 2020, with some reporting significantly lower symptom scores than strains without mutations, whereas others reported no significant differences in clinical presentations [60,61]. However, because this study included only outpatients, and most patients had relatively mild symptoms, the association between mutation and severity could not be confirmed. In addition, phylogenetic tree analysis detected strains with D214Y and G272S mutations in RSV-A and an H128R mutation in RSV-B for the first time after the COVID-19 pandemic. Furthermore, these mutations were not found in strains from other countries, which were considered to be more than 98% closely related by BLAST. G proteins are prone to mutation because they are under selective pressure for host immunity [10]. While the functions are yet to be elucidated, the appearance of these mutations indicates that RSV may have evolved locally during a period of restricted social activity as overseas flights were restricted to Japan in 2020, and it is unlikely that new RSV strains were introduced from outside the country. Similar events were reported in Taiwan and China [32,57]. However, further studies are required to determine the clinical or virologic changes that these local evolutions may induce.

Longitudinal phylogeographic analysis of HVR2 in the G gene revealed interesting findings over the 10 years. At least seven clusters of ON1 in RSV-A circulated with possible roots from populated areas, such as Mie (close to Nagoya), Shiga (close to Kyoto and Osaka), and Tokyo, but decreased to two after the pandemic by 2022, from the strains conserved in Okinawa. This suggests that several clusters may have been eliminated because of the NPIs in 2020. A similar phenomenon was reported in Australia in late 2020 [31]. In contrast, this study showed a monophyletic evolution of RSV-B. Interestingly, we found that Okinawa was the source of the revival of RSV-B after the pandemic, as observed for RSV-A. Ono et al. reported the polyphyletic evolution of RSV-A and monophyletic evolution of RSV-B in a local area in Japan during the pandemic [38]. Thus, our study had the unique advantage of covering multiple areas of Japan to observe the geographic transmission of RSV-A and -B before and after the pandemic.

This study has some limitations. First, sampling was biased by year and prefecture because our collaborating physicians were situated as one doctor in one prefecture, making it difficult to use the data qualitatively. Second, since 2021, the number of facilities participating in surveillance has decreased from 11 to 8. After 2020, the participating clinicians were overburdened by treating patients with COVID-19, which made it difficult for them to participate in our surveillance and conduct sampling. To achieve a more accurate understanding of the prevalence in each prefecture and enhance phylogeographic analysis, a more robust sampling approach should have been implemented. Ensuring consistent and geographically representative sampling throughout Japan was crucial. This limitation affects the reliability of the phylogeographic analysis, given its sensitivity to the number of strains and the timing of sampling in each prefecture. Third, phylogeographic analysis was not performed on the G ectodomain region throughout 2012–2022 since the samples collected before 2018 were sequenced only for HVR2. However, HVR2 has long been used in the molecular genotyping analysis of RSV by various researchers, allowing for comparison with previous studies. Fourth, this study analyzed only the G protein and not the whole genome. Recently, whole-genome analysis using next-generation sequencing has emerged as the main method for RSV genotyping [62]. Whole-genome analysis enables the coverage of the F protein, which is the target of palivizumab [63], and RSV vaccines that were recently approved [4,5,6].

## 5. Conclusions

This study showed that although there were changes in the epidemic seasons of RSV in Japan, the main genotypes of RSV-A and -B did not change before and after the COVID-19 pandemic in Japan. Although the diversity of RSV-A clusters decreased after the COVID-19 pandemic, RSV-B retained its monophyletic evolution in Japan for approximately 10 years. Despite fluctuating data across prefectures, and a lack of research covering various areas for RSV sampling in Japan except for our group before and after the COVID-19 pandemic, our study yields valuable findings on the surviving genotypes of RSV in Japan. It remains to be seen whether RSV seasonality will return to what was observed in the pre-pandemic period and whether the diversity of RSV strains will thrive again.

## Figures and Tables

**Figure 1 viruses-15-02382-f001:**
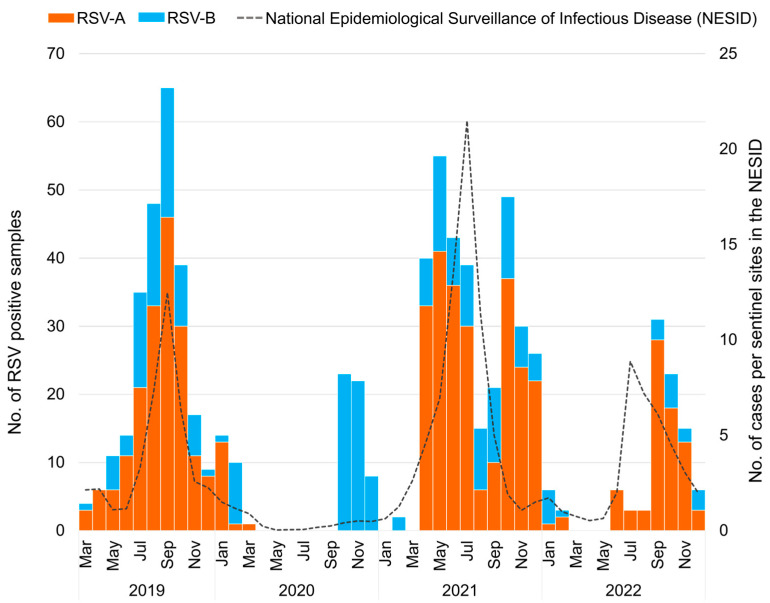
The monthly distribution of respiratory syncytial virus (RSV) collected in Japan between March 2019 and December 2022 (*n* = 739). The orange bar graph represents the number of RSV-A-positive samples detected using real-time polymerase chain reaction (RT-PCR); the blue bar graph represents the number of RSV-B-positive samples detected using RT-PCR; the dashed line represents the number of cases in the National Epidemiological Surveillance of Infectious Disease (NESID) reported from the National Institute of Infectious Diseases (NIID).

**Figure 2 viruses-15-02382-f002:**
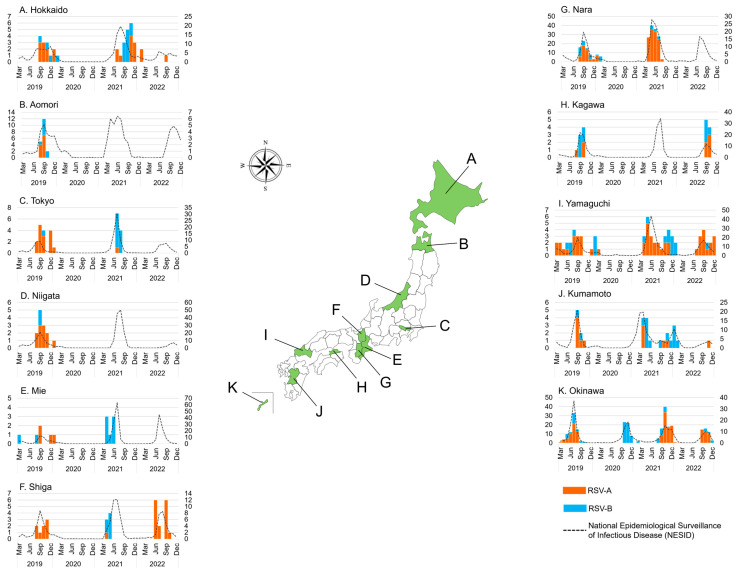
The monthly distribution of collected respiratory syncytial virus (RSV) in 11 Japanese prefectures between March 2019 and December 2022 (*n* = 739). The orange bar graph represents the number of RSV-A-positive samples detected using real-time polymerase chain reaction (RT-PCR); the blue bar graph represents the number of RSV-B-positive samples detected using RT-PCR; the dashed line represents the number of cases in the National Epidemiological Surveillance of Infectious Disease (NESID) by the National Institute of Infectious Diseases (NIID). The left *y*-axis shows the number of RSV-positive samples in this study, and the right *y*-axis shows the number of RSV cases in the NESID.

**Figure 3 viruses-15-02382-f003:**
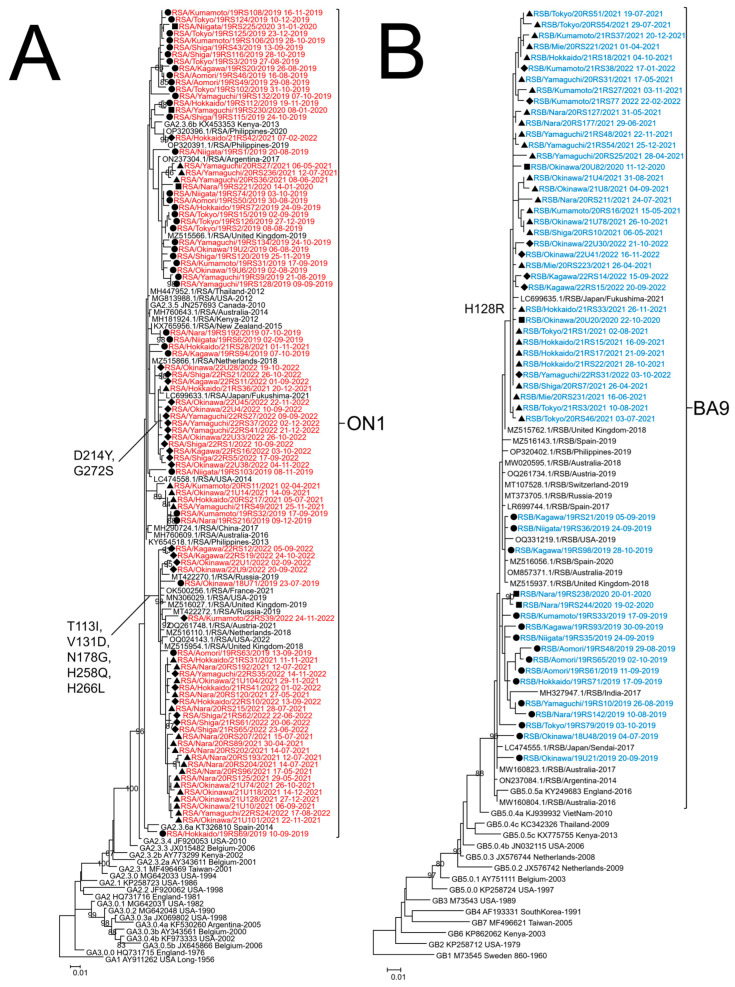
Phylogenetic tree analysis of respiratory syncytial virus (RSV) strains collected in Japan from 2019 to 2022. Maximum likelihood (ML) phylogenetic trees of glycoprotein (G) ectodomain region gene of RSV-A (**A**) and RSV-B (**B**) were generated by MEGA version 6 software. Strains detected in 2019 are indicated by a closed circle (●), those detected in 2020 are indicated by a closed square (■), those detected in 2021 are indicated by a closed triangle (▲), and those detected in 2022 are indicated by a closed diamond (◆). RSV-A strains obtained in this study are colored red, RSV-B strains are colored blue, and reference strains from Japan and the other countries are colored black. The strains obtained in this study are named using the RSA or RSB/collected prefecture/number of specimen/collection year collection date format. Only bootstrap values ≥ 80% are shown at the branch nodes. The length of the branch represents the evolutionary distance along the evolutionary tree. The genotypes are shown on the right square brackets.

**Figure 4 viruses-15-02382-f004:**
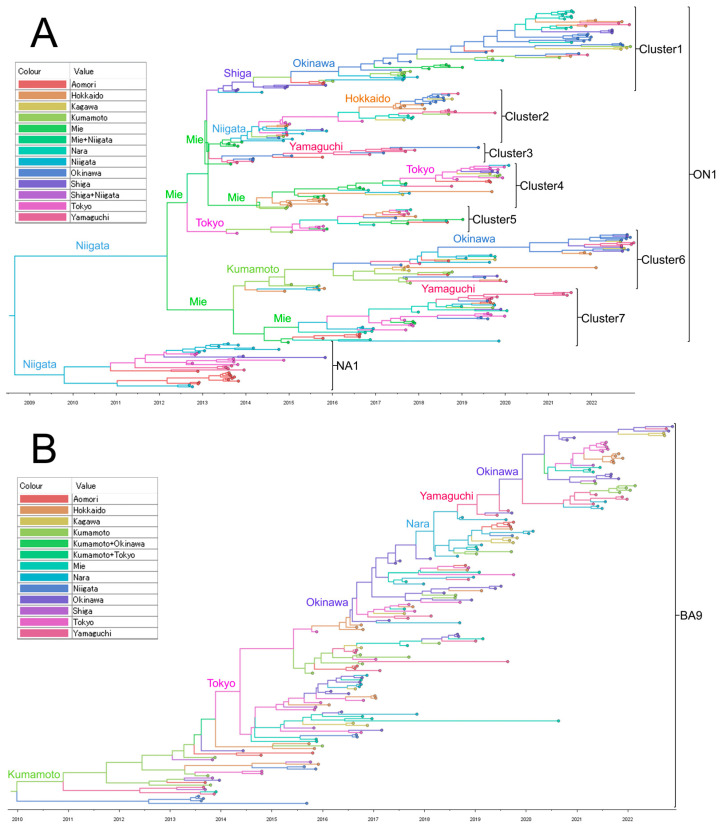
Time scale maximum clade credibility (MCC) tree of respiratory syncytial virus (RSV) glycoprotein (G) second hypervariable region (HVR2) of RSV-A (**A**) and RSV-B (**B**). These MCC trees were generated using the Markov Chain Monte Carlo (MCMC) method. The branches are in a time scale in years and are color-coded according to the location of the most probable ancestor of descendant nodes. The genotypes are shown on the right square brackets.

**Table 1 viruses-15-02382-t001:** Characteristics of patients with respiratory syncytial virus (RSV) detected using real-time polymerase chain reaction (RT-PCR) in Japan between March 2019 and December 2022 (*n* = 739).

Variable	2019, *n* (%)	2020, *n* (%)	2021, *n* (%)	2022, *n* (%)	Total, *n* (%)	*p*-Value
Total number of patients	248 (100)	78 (100)	320 (100)	93 (100)	739 (100)	
Age (years), median [IQR]	1.3 [0.8–1.9]	1.2 [0.6–1.7]	1.7 [1.0–2.6]	1.4 [1.0–2.2]	1.4 [0.8–2.2]	
Age group						<0.001
0 to <2	186 (75.0)	64 (82.1)	188 (58.8)	65 (69.9)	503 (68.1)	
2 to <6	62 (25.0)	14 (17.9)	132 (41.3)	28 (30.1)	236 (31.9)	
Sex						0.151
Male	123 (49.6)	38 (48.7)	186 (58.1)	47 (50.5)	394 (53.3)	
Female	125 (50.4)	40 (51.3)	134 (41.9)	46 (49.5)	345 (46.7)	
Symptoms						
Fever	207 (83.5)	60 (76.9)	265 (82.8)	71 (76.3)	603 (81.6)	0.281
Cough	245 (98.8)	76 (97.4)	312 (97.5)	87 (93.5)	720 (97.4)	0.068
Rhinorrhea	245 (98.8)	76 (97.4)	305 (95.3)	83 (89.2)	709 (95.9)	0.001
Premature birth or Underlying Disease						0.739
Yes	20 (8.1)	7 (9.0)	17 (5.3)	7 (7.5)	51 (6.9)	
No	228 (91.9)	71 (91.0)	303 (94.7)	86 (92.5)	688 (93.1)	

Abbreviations: IQR, interquartile range. Significant differences were found between age: 2019 and age: 2021 (*p* < 0.001), age: 2020 and age: 2021 (*p* < 0.001), Rhinorrhea: 2019 and Rhinorrhea: 2021 (*p* = 0.027), Rhinorrhea: 2019 and Rhinorrhea: 2022 (*p* < 0.001), and Rhinorrhea: 2021 and Rhinorrhea: 2022 (*p* = 0.045).

**Table 2 viruses-15-02382-t002:** Subgroup of respiratory syncytial virus (RSV) divided by prefecture between March 2019 and December 2022 (*n* = 967).

Prefecture	2019, *n* (%)				2020, *n* (%)				2021, *n* (%)				2022, *n* (%)			
	No. of CS	No. of RT-PCR POS	RSV -A	RSV -B	No. of CS	No. of RT-PCR POS	RSV -A	RSV -B	No. of CS	No. of RT-PCR POS	RSV -A	RSV -B	No. of CS	No. of RT-PCR POS	RSV -A	RSV -B
Hokkaido	26 (100)	11 (42.3)	8 (30.8)	3 (11.5)	5 (100)	3 (60.0)	2 (40.0)	1 (20.0)	22 (100)	20 (90.9)	10 (45.5)	10 (45.5)	3 (100)	3 (100)	3 (100)	0 (0.0)
Aomori	21 (100)	19 (90.5)	11 (52.4)	8 (38.1)	-	-	-	-	-	-	-	-	-	-	-	-
Tokyo	19 (100)	15 (78.9)	14 (73.7)	1 (5.3)	1 (100)	1 (100)	1 (100)	0 (0.0)	13 (100)	11 (84.6)	1 (7.7)	10 (76.9)	-	-	-	-
Niigata	15 (100)	12 (80.0)	10 (66.7)	2 (13.3)	1 (100)	1 (100)	1 (100)	0 (0.0)	3 (100)	0 (0.0)	0 (0.0)	0 (0.0)	-	-	-	-
Mie	5 (100)	5 (100)	3 (60.0)	2 (40.0)	1 (100)	1 (100)	1 (100)	0 (0.0)	10 (100)	7 (70.0)	0 (0.0)	7 (70.0)	-	-	-	-
Shiga	14 (100)	8 (57.1)	8 (57.1)	0 (0.0)	-	-	-	-	13 (100)	7 (53.8)	1 (7.7)	6 (46.2)	29 (100)	12 (41.4)	12 (41.4)	0 (0.0)
Nara	81 (100)	65 (80.2)	43 (53.1)	22 (27.2)	15 (100)	14 (93.3)	8 (53.3)	6 (40.0)	164 (100)	134 (81.7)	124 (75.6)	10 (6.1)	-	-	-	-
Kagawa	17 (100)	8 (47.1)	3 (17.6)	5 (29.4)	-	-	-	-	-	-	-	-	9 (100)	9 (100)	5 (55.6)	4 (44.4)
Yamaguchi	21 (100)	19 (90.5)	15 (71.4)	4 (19.0)	6 (100)	5 (83.3)	2 (33.3)	3 (50.0)	36 (100)	27 (75.0)	19 (52.8)	8 (22.2)	22 (100)	17 (77.3)	14 (63.6)	3 (13.6)
Kumamoto	14 (100)	8 (57.1)	6 (42.9)	2 (14.3)	-	-	-	-	19 (100)	14 (73.7)	5 (26.3)	9 (47.4)	5 (100)	5 (100)	1 (20.0)	4 (80.0)
Okinawa	86 (100)	78 (90.7)	54 (62.8)	24 (27.9)	90 (100)	53 (58.9)	0 (0.0)	53 (58.9)	132 (100)	100 (75.8)	79 (59.8)	21 (15.9)	49 (100)	47 (95.9)	39 (79.6)	8 (16.3)
All	319 (100)	248 (77.7)	175 (54.9)	73 (22.9)	119 (100)	78 (65.5)	15 (12.6)	63 (52.9)	412 (100)	320 (77.7)	239 (58.0)	81 (19.7)	117 (100)	93 (79.5)	74 (63.2)	19 (16.2)

Abbreviations: CS: collected samples; RT-PCR: real-time polymerase chain reaction; POS: positive samples; RSV: respiratory syncytial virus. Note: “-” denotes samples were not collected.

## Data Availability

The glycoprotein (G) ectodomain nucleotide sequences were registered in the GISAID database under the accession numbers EPI_ISL_15953175 to EPI_ISL_18487961. The HVR2 nucleotide sequences collected from 2015 to 2018 were registered in the GISAID database under the accession numbers EPI_ISL_18396054 to EPI_ISL_18406067.

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
