# Peer review of "Molecular Epidemiology of Respiratory Syncytial Virus during 2019–2022 and Surviving Genotypes after the COVID-19 Pandemic in Japan"

_viruses, 2023, doi:10.3390/v15122382_

Round 1

Reviewer 1 Report

Comments and Suggestions for Authors

The manuscript authored by Yoshioka and colleagues presents findings on molecular epidemiology of respiratory syncytial virus during 2019-2022 and surviving genotypes after the COVID-19 pandemic in Japan.

Therefore, this manuscript by Yoshioka et al. has the major findings: 

“1) Of 967 samples, 739 (76.4%) were found to be RSV positive using RT-PCR. In addition, RSV peaked in September 2019 but was not detected in 2020, except in Okinawa. Nationwide epidemics occurred with peaks in July 2021 and 2022; 

2) The genotype remained the same, ON1 for RSV-A and BA9 for RSV-B during 2019–2022. Phylogeographic analysis of HVR revealed that at least seven clusters of RSV-A had circulated previously but decreased to two clusters after the pandemic, whereas RSV-B had a single monophyletic cluster over 10 years."

From a scientific standpoint, the subject tackled by the authors is pertinent and encapsulates a contemporary health concern pertaining to influenza-like illness cases. Therefore, I didn't identify any issues with this manuscript. The approach, analysis, and presentation of results are well-done.

 I have minor points that I see interesting to add to this manuscript:

Table 2: Use abbreviations to improve table layout (i.e., no. RT-PCR POS1; no. CS2). Abbreviations: 1Positive; 2Collected samples.

Reviewer 2 Report

Comments and Suggestions for Authors

Manuscript: "Molecular Epidemiology of Respiratory Syncytial Virus During 2019–2022…” by Yoshioka et al. 

This is very likely the most detailed survey and analysis of all patient isolates from various Prefectures in Japan for their various RSV strain and variant content, covering the two established serotypes, A and B. The authors have carried out an exemplary analysis of RSV G gene sequences, phylogeny and state-of-the-art biostatistical analyses. In fact the study enlisted an impressive number of collaborators spanning multiple laboratories from Hokkaido in the North to Kumamoto and Okinawa in the South. The period covers the critical years 2019-2022, spanning the peak of the CoVID-19 pandemic and after. The main focus is the 2-<6 year old children, the most vulnerable age group that also develops susceptibility to allergens and asthma later in life as a result of childhood RSV infection. 

My main concern is that although the RSV genotype analysis strong and comprehensive, it did not provide anything useful or surprising. There are variations and data scatter between and within the multiple prefectures, for which no clear mechanism could be established. The original premise to reveal any effect of CoVID-19 on RSV sounded interesting, but did not reveal much at the end. There was no evidence that seasonal peaks were affected by SARS-CoV-2 infection, immunity, or vaccination. The authors do not offer a reason for the 60-nt duplication in RSV-B or the H128R mutation. Why is rhinorrhea lower in 2021, 2022 than in 2019 etc? Thus, the conclusions have remained a plain summary of the results, without a mechanistic discovery that could be useful to scientists and readers outside Japan.  

Minor comments, mostly typographical errors:

Line 57: should be ‘Pneumoviridae’

Line 59: should be 15 kb

Line 98: should be ‘located in 11 different prefectures..’

Less important: The article ‘the’ is sometimes missing.

Comments on the Quality of English Language

Already listed in the comment to the authors.
